# An Efficient Quantum Secret Sharing Scheme Based on Restricted Threshold Access Structure

**DOI:** 10.3390/e25020265

**Published:** 2023-01-31

**Authors:** Lei Li, Zhi Li

**Affiliations:** School of Mechano-Electronic Engineering, Xidian University, Xi’an 710071, China

**Keywords:** quantum secret sharing, phase shift operation, GHZ state, efficiency

## Abstract

Quantum secret sharing is an important branch of quantum cryptography, and secure multi-party quantum key distribution protocols can be constructed using quantum secret sharing. In this paper, we construct a quantum secret sharing scheme built on a constrained (*t*, *n* ) threshold access structure, where *n* is the number of participants and *t* is the threshold number of participants and the distributor. Participants from two different sets perform the corresponding phase shift operations on two particles in the GHZ state passed to them, and then t−1 participants with the distributor can recover the key, where the participant recovering the key measures the particles received by himself and finally obtains the key through the collaboration of the distributors. Security analysis shows that this protocol can be resistant to direct measurement attacks, interception retransmission attacks, and entanglement measurement attacks. This protocol is more secure, flexible, and efficient compared with similar existing protocols, which can save more quantum resources.

## 1. Introduction

Secret sharing is an important branch of information security research, and it provides new ideas for solving key management problems [1,2]. The secret sharing system divides the shared secrets into sub-secrets, which are sent separately to several participants for safekeeping, and it specifies which participants can restore the secrets together and which participants cannot cooperate to obtain the approved secret information. Quantum secret sharing is an important research direction in quantum cryptography, which combines quantum theory and classical secret sharing and belongs to a kind of quantum key distribution in quantum key management [3,4,5,6].

Quantum secret sharing means that the distributor divides a classical or quantum message into several copies, and only the participants in the authorized set can recover the secret, while the participants in the non-authorized set cannot recover the secret. The security of the quantum secret sharing scheme is significantly improved in terms of the security of sharing compared to computationally complex classical secrets due to the guarantee of the relevant fundamental principles in quantum exploitation.

The first quantum secret sharing (QSS) scheme was proposed by Hillery [7] in 1999 using the Greenberger–Horne–Zeilinger (GHZ) state. In the same year, Cleve et al. [6] proposed the threshold QSS scheme using the quantum error correction code theory, where the threshold quantum secret sharing scheme means that the distributor divides the secret information into *n* copies and sends them to *n* participants separately, and at least *t* participants cooperate to recover the secret; however, the set of less than *t* participants cannot recover the secret. Since then, increasingly quantum secret sharing schemes have been proposed [8,9,10,11,12,13,14], and some of these schemes are based on quantum physical properties to share classical information, while some schemes are based on quantum mechanics principles to share arbitrary quantum state information.

Many researchers have designed series of different types of schemes based on different quantum principles, such as based on single photons [15,16,17], product states [18,19,20], and entangled states [21,22,23,24,25], respectively. Among the above secret sharing schemes, threshold schemes occupy an important position [6,13,26,27,28,29,30,31]; however, in practical applications, the authorized subset may not consist of any *t* participants, and there are some occasions in which the permissions of certain participants are restricted, such as confidential data restoration, hierarchical structures, and financial infidelity. Therefore, the (t,n) threshold structure is not suitable for these occasions.

In 2013, Gheorghiu et al. [21] constructed the first quantum secret sharing scheme by local operations and classical communication (LOCC); in 2015, Rahaman et al. [22] gave a QSS model based on LOCC. The above two schemes are built on restricted (t,n) threshold type access structures. This type of access structure does not belong to the general (t,n) threshold structure and can satisfy the use of secret sharing in some special cases. Since this scheme is simple and efficient [22], a number of scholars have constructed many QSS schemes based on this class of restricted access structures on the basis of this property of local distinguishability of quantum states [23,24,25,26].

However, all the above schemes utilize the entangled states of *n* particles, and when the number of participants *n* is large, *n*-qudit entangled states are currently difficult to make. Therefore, how to utilize the entangled states of a small number of particles for such restricted threshold structures to accomplish the distribution of multi-party quantum keys is a problem that needs to be solved in the construction of QSS schemes. In this paper, we use phase shift operation based on three-particle entangled states to achieve multi-party quantum key distribution on this kind of restricted access structures, which is an efficient and secure protocol, and at the same time, saves more quantum resources compared with similar protocols.

This paper is organized as follows: In Section 1, we give the phase shift operator and its properties. Then, the detailed procedure of the scheme is given in Section 2. Next, the correctness and security of this scheme is proven in Section 3 and Section 4, respectively. The efficiency and other metrics of this protocol are compared with several protocols of the same type in Section 5. Finally, a short conclusion is provided in Section 6.

## 2. Preliminary Knowledge

This section further studies the properties of unitary operators on the basis of literature [28], providing a theoretical basis for constructing the multi-party quantum key distribution protocol in this paper. Let $Z_{p}$ be a finite field and *p* be an odd prime number. The GHZ states used in this paper are $|{\mathrm{GHZ}}_{000}\rangle$ and $|{\mathrm{GHZ}}_{100}\rangle$, where
$$\begin{array}{c} |{\mathrm{GHZ}}_{000}\rangle =\frac{1}{\sqrt{2}}(|000\rangle +\left|111\rangle ),\right|{\mathrm{GHZ}}_{100}\rangle =\frac{1}{\sqrt{2}}(|000\rangle -|111\rangle ). \end{array}$$

An angle *a* shift operation is performed on the relative phase on the *j*-TH particle of GHZ, denoting by $U_{j}\left(a\right)$, where
$$\begin{array}{c} U_{j}\left(a\right)=\left(\begin{array}{cc} 1 & 0\\ 0 & e^{i\cdot a} \end{array}\right), \end{array}$$
where $a\in Z_{p},j=1,2,3$.

**Lemma** **1.**
*For the |GHZ〉 state, we have*

(1)
$$\begin{array}{c} U_{1}\left(a\right)⊗I⊗I|\mathrm{GHZ}\rangle \\ =I⊗U_{2}\left(a\right)⊗I|\mathrm{GHZ}\rangle \\ =I⊗I⊗U_{3}\left(a\right)|\mathrm{GHZ}\rangle , \end{array}$$

*where I is a constant operator.*


**Proof.** Prove that the equation holds only for the case $\left|\mathrm{GHZ}\rangle =\right|{\mathrm{GHZ}}_{000}\rangle$. Other cases can be proven similarly.
$$\begin{array}{c} \begin{array}{cc} U_{1}\left(a\right)⊗I⊗I|\mathrm{GHZ}\rangle & =U_{1}\left(a\right)⊗I⊗I\left[\frac{1}{\sqrt{2}}(|000\rangle +\left|111\rangle \right)\right]\\ & =\frac{1}{\sqrt{2}}\left[\left(U_{1}\left(a\right)|0\rangle ⊗|00\rangle +U_{1}\left(a\right)|1\rangle ⊗|11\rangle \right)\right]\\ & \left.=\frac{1}{\sqrt{2}}[|0\rangle ⊗\left|00>+e^{i\cdot a}\right|1\rangle ⊗|11\rangle \right]\\ & =\frac{1}{\sqrt{2}}\left(|000\rangle +e^{i\cdot a}|111\rangle \right). \end{array} \end{array}$$Similarly, it can be proven that
$$\begin{array}{c} \begin{array}{c} I⊗U_{2}\left(a\right)⊗I|\mathrm{GHZ}\rangle =\frac{1}{\sqrt{2}}\left(|000\rangle +e^{i\cdot a}|111\rangle \right).\\ I⊗I⊗U_{3}\left(a\right)|\mathrm{GHZ}\rangle =\frac{1}{\sqrt{2}}\left(|000\rangle +e^{i\cdot a}|111\rangle \right). \end{array} \end{array}$$Thus, Lemma 1 holds when $\left|\mathrm{GHZ}\rangle =\right|{\mathrm{GHZ}}_{100}\rangle$. □

Lemma 1 shows the result that a shift of angle *a* to particle 1 of the GHZ state is equivalent to a shift of angle *a* to its particle 2 or 3.

**Lemma** **2.**
*For the |GHZ〉 state, we have*

(2)
$$\begin{array}{c} \left(U_{1}\left(a\right)⊗I⊗I\right)\left(U_{1}\left(b\right)⊗I⊗I\right)|\mathrm{GHZ}\rangle =U_{1}\left(a+b\right)⊗I⊗I|\mathrm{GHZ}\rangle ; \end{array}$$


(3)
$$\begin{array}{c} I⊗\left(U_{2}\left(a\right)⊗I\right)\left(I⊗U_{2}\left(b\right)⊗I\right)|\mathrm{GHZ}\rangle =I⊗U_{2}\left(a+b\right)⊗I|\mathrm{GHZ}\rangle ; \end{array}$$


(4)
$$\begin{array}{c} I⊗I⊗\left(U_{3}\left(a\right)\right)\left(I⊗I⊗U_{3}\left(b\right)\right)|\mathrm{GHZ}\rangle =I⊗I⊗U_{3}\left(a+b\right)|\mathrm{GHZ}\rangle . \end{array}$$



**Proof.** We only prove that the equation holds for the case of $\left|\mathrm{GHZ}\rangle =\right|{\mathrm{GHZ}}_{000}\rangle$; the other cases can be proven similarly. Since
$$\begin{array}{c} \begin{array}{cc} \left(U_{1}\left(a\right)⊗I⊗I\right) & \left(U_{1}\left(b\right)⊗I⊗I\right)|\mathrm{GHZ}\rangle =U_{1}\left(a+b\right)⊗I⊗I|\mathrm{GHZ}\rangle \\ & U_{1}\left(a+b\right)⊗I⊗I|\mathrm{GHZ}\rangle =U_{1}\left(a+b\right)⊗I⊗I\left[\frac{1}{\sqrt{2}}(|000\rangle +\left|111\rangle \right)\right]\\ & =\frac{1}{\sqrt{2}}\left[U_{1}\left(a+b\right)|0\rangle ⊗|00\rangle +U_{1}\left(a+b\right)|1\rangle ⊗|11\rangle \right]\\ & =\frac{1}{\sqrt{2}}\left[|0\rangle ⊗|00\rangle +e^{i\cdot (a+b)}|1\rangle ⊗|11\rangle \right].\\ & =\frac{1}{\sqrt{2}}\left[|000\rangle +e^{i\cdot (a+b)}|111\rangle \right] \end{array} \end{array}$$
and
$$\begin{array}{c} \begin{array}{c} \left(U_{1}\left(a\right)⊗I⊗I\right)\left(U_{1}\left(b\right)⊗I⊗I\right)|\mathrm{GHZ}\rangle \\ =\left(U_{1}\left(a\right)⊗I⊗I\right)\left(U_{1}\left(b\right)⊗I⊗I\right)\left[\frac{1}{\sqrt{2}}(|000\rangle +\left|111\rangle \right)\right]\\ \left.=\frac{1}{\sqrt{2}}U_{1}\left(a\right)⊗I⊗I\right)\left[U_{1}\left(b\right)|0\rangle ⊗|00\rangle +U_{1}\left(b\right)|1\rangle ⊗|11\rangle \right]\\ \left.=\frac{1}{\sqrt{2}}U_{1}\left(a\right)⊗I⊗I\right)\left[|0\rangle ⊗|00\rangle +e^{i\cdot b}|1\rangle ⊗|11\rangle \right]\\ =\frac{1}{\sqrt{2}}\left[U_{1}\left(a\right)|0\rangle ⊗|00\rangle +e^{i\cdot b}U_{1}\left(a\right)|1\rangle ⊗|11\rangle \right]\\ =\frac{1}{\sqrt{2}}\left[|0\rangle ⊗|00\rangle +e^{i\cdot b}e^{ia}|1\rangle ⊗|11\rangle \right]\\ \left.=\frac{1}{\sqrt{2}}\left[|000\rangle +e^{i\cdot (a+b)}|111\rangle \right)\right] \end{array} \end{array}$$Therefore, when $\left|\mathrm{GHZ}\rangle =\right|{\mathrm{GHZ}}_{000}\rangle$, $\left(U_{1}\left(a\right)⊗I⊗I\right)\left(U_{1}\left(b\right)⊗I⊗I\right)|\mathrm{GHZ}\rangle =U_{1}\left(a+b\right)⊗I⊗I|\mathrm{GHZ}\rangle$ holds. It can be proven in the same way that, when $\left|\mathrm{GHZ}\rangle =\right|{\mathrm{GHZ}}_{100}\rangle$, there is $\left(U_{1}\left(a\right)⊗I⊗I\right)\left(U_{1}\left(b\right)⊗I⊗I\right)|\mathrm{GHZ}\rangle =U_{1}\left(a+b\right)⊗I⊗I|\mathrm{GHZ}\rangle$. □

Lemma 2 shows that the result of performing two consecutive shifts of angles *a* and *b* on a particle of the quantum state GHZ is equivalent to performing a shift of angle a+b on this particle. Using Lemma 2, by induction, we have the following result:

**Theorem** **1.**
*For the |GHZ〉 state, we have,*

(5)
$$\begin{array}{c} \left(U_{1}\left(a_{1}\right)⊗I⊗I\right)\cdots \left(U_{1}\left(a_{l}\right)⊗I⊗I\right)|\mathrm{GHZ}\rangle =U_{1}\left(a_{1}+\cdots +a_{l}\right)⊗I⊗I|\mathrm{GHZ}\rangle ; \end{array}$$


(6)
$$\begin{array}{c} \left(I⊗U_{2}\left(a_{1}\right)⊗I\right)\cdots \left(I⊗U_{2}\left(a_{l}\right)⊗I\right)|\mathrm{GHZ}\rangle =I⊗U_{2}\left(a_{1}+\cdots +a_{l}\right)⊗I|\mathrm{GHZ}\rangle ; \end{array}$$


(7)
$$\begin{array}{c} \left(I⊗I⊗U_{3}\left(a_{1}\right)\right)\cdots \left(I⊗I⊗U_{3}\left(a_{l}\right)\right)|\mathrm{GHZ}\rangle =I⊗I⊗U_{3}\left(a_{1}+\cdots +a_{l}\right)|\mathrm{GHZ}\rangle . \end{array}$$



Theorem 1 shows that the result of performing *l* successive shifts of angle $a_{i}$ on a particle of the quantum state |GHZ〉 is equivalent to performing a shift of angle $a_{1}+a_{2}+\cdots +a_{l}$ on this particle, where i=1,2,⋯,l.

**Theorem** **2.**
*For the |GHZ〉 state, we have,*

(8)
$$\begin{array}{c} U_{1}\left(a_{1}\right)⊗U_{2}\left(a_{2}\right)⊗U_{3}\left(a_{3}\right)|\mathrm{GHZ}\rangle =U_{1}\left(a_{1}+a_{2}+a_{3}\right)⊗I⊗I|\mathrm{GHZ}\rangle ; \end{array}$$


(9)
$$\begin{array}{c} U_{1}\left(a_{1}\right)⊗U_{2}\left(a_{2}\right)⊗U_{3}\left(a_{3}\right)|\mathrm{GHZ}\rangle =I⊗U_{2}\left(a_{1}+a_{2}+a_{3}\right)⊗I|\mathrm{GHZ}\rangle ; \end{array}$$


(10)
$$\begin{array}{c} U_{1}\left(a_{1}\right)⊗U_{2}\left(a_{2}\right)⊗U_{3}\left(a_{3}\right)|\mathrm{GHZ}\rangle =I⊗I⊗U_{3}\left(a_{1}+a_{2}+a_{3}\right)|\mathrm{GHZ}\rangle . \end{array}$$



**Proof.** Prove that the equation holds for the case of $\left|\mathrm{GHZ}\rangle =\right|{\mathrm{GHZ}}_{000}\rangle$ only. The other cases can be proven similarly. First, prove that Equation (Equation 8) holds. Since
(11)$$\begin{array}{c} U_{1}\left(a_{1}\right)⊗U_{1}\left(a_{2}\right)⊗U_{3}\left(a_{3}\right)|\mathrm{GHZ}\rangle \\ =\left(U_{1}\left(a_{1}\right)⊗I⊗I\right)\left(I⊗U_{1}\left(a_{2}\right)⊗I\right)\left(I⊗I⊗U_{3}\left(a_{3}\right)\right)|\mathrm{GHZ}\rangle \\ =\left(U_{1}\left(a_{1}\right)⊗I⊗I\right)\left(I⊗U_{1}\left(a_{2}\right)⊗I\right)\frac{1}{\sqrt{2}}\left(|000\rangle +e^{i\cdot a_{3}}|111\rangle \right)\\ =\left(U_{1}\left(a_{1}\right)⊗I⊗I\right)\frac{1}{\sqrt{2}}\left(|000\rangle +e^{i\cdot a_{3}}e^{i\cdot a_{2}}|111\rangle \right)\\ =\frac{1}{\sqrt{2}}\left(|000\rangle +e^{i\cdot \left(a_{3}+a_{2}\right)}e^{i\cdot a_{1}}|111\rangle \right)\\ =\frac{1}{\sqrt{2}}\left(|000\rangle +e^{i\cdot \left(a_{3}+a_{2}+a_{1}\right)}|111\rangle \right)\\ =U_{1}\left(a_{1}+a_{2}+a_{3}\right)⊗I⊗I|\mathrm{GHZ}\rangle . \end{array}$$Then, $U_{1}\left(a_{1}\right)⊗U_{2}\left(a_{2}\right)⊗U_{3}\left(a_{3}\right)|\mathrm{GHZ}\rangle =U_{1}\left(a_{1}+a_{2}+a_{3}\right)⊗I⊗I|\mathrm{GHZ}\rangle$; therefore, (8) holds.On the other hand, from Lemma 1, we have
$$\begin{array}{c} \begin{array}{c} U_{1}\left(a_{1}+a_{2}+a_{3}\right)⊗I⊗I|\mathrm{GHZ}\rangle \\ =I⊗U_{2}\left(a_{1}+a_{2}+a_{3}\right)⊗I|\mathrm{GHZ}\rangle \\ =I⊗I⊗U_{3}\left(a_{1}+a_{2}+a_{3}\right)|\mathrm{GHZ}\rangle . \end{array} \end{array}$$Combining Equation (Equation 11) gives
$$\begin{array}{c} \begin{array}{c} U_{1}\left(a_{1}\right)⊗U_{2}\left(a_{2}\right)⊗U_{3}\left(a_{3}\right)|\mathrm{GHZ}\rangle =I⊗U_{2}\left(a_{1}+a_{2}+a_{3}\right)⊗I|\mathrm{GHZ}\rangle ,\\ U_{1}\left(a_{1}\right)⊗U_{2}\left(a_{2}\right)⊗U_{3}\left(a_{3}\right)|\mathrm{GHZ}\rangle =I⊗I⊗U_{3}\left(a_{1}+a_{2}+a_{3}\right)|\mathrm{GHZ}\rangle . \end{array} \end{array}$$Therefore, Equations (9) and (10) hold.Similarly, it follows that, when $\left|\mathrm{GHZ}\rangle =\right|{\mathrm{GHZ}}_{000}\rangle$ holds, then (8), (9), and (10) hold. □

## 3. The Proposed Protocol

In this section, we propose a multi-party quantum secret sharing protocol based on generalized GHZ states. This QSS protocol is divided into three phases: the initial phase, share distribution phase, and secret reconstruction phase.

### 3.1. Initialization Phase

Let *P* be a set containing *n* participants with $P=\left\{P_{1},P_{2},\cdots ,P_{n}\right\}$. Let $P^{\left(1\right)}=\{P_{1}^{\left(1\right)},\cdots ,$$P_{t_{1}}^{\left(1\right)}\}$ and $P^{\left(2\right)}=\left\{P_{1}^{\left(2\right)},\cdots ,P_{t_{2}}^{\left(2\right)}\right\}$ be, respectively, two subsets of *P*, where $t_{i}\geq 1$ and satisfies $t_{1}+t_{2}=t-1,3\leq t\leq n$. The distributor Alice chooses a prime d(2<d<2n) and sets a finite field $Z_{d}$. Alice then chooses a (t−1)-degree polynomial $f\left(x\right)=S+a_{1}x^{1}+\cdots +a_{t-1}x^{t-1}$, where *S* is a secret, $f\left(x\right)\in Z_{d}\left[x\right]$, and the symbol ${}^{‘}{+}^{’}$ is defined as the modulo addition. Let $m=\left\lceil \log_{2}d\right\rceil$, and then *S* can be represented as a binary sequence, i.e., $S=\left(s_{1},s_{2},\cdots ,s_{m}\right)$, where $s_{i}\in \left\{0,1\right\}$, i=1,2,⋯,m. Alice chooses the SHA1 hash function H(S) with key and computes H(S), then shares H(S) with the participants from the set *P*.

### 3.2. Share Distribution Phase

In this phase, Alice shares sub-shares among the participants in the set $P^{\left(1\right)}\cup P^{\left(2\right)}$.


**(1) Distribution of classic shares**


Alice computes the classical share $f\left(x_{r}^{\left(j\right)}\right)$ and assigns $f\left(x_{r}^{\left(j\right)}\right)$ to the participant $P_{r}^{\left(j\right)}$ via a secure channel (e.g., a quantum direct communication channel), and Alice computes her own share $f(x_{0})$ as well as $S_{0}=f\left(x_{0}\right)\prod_{1\leq r\leq t_{1}}\frac{x_{r}^{\left(1\right)}}{\left(x_{r}^{\left(1\right)}-x_{0}\right)}\prod_{1\leq r\leq t_{2}}\frac{x_{r}^{\left(2\right)}}{\left(x_{r}^{\left(2\right)}-x_{0}\right)}$, where $x_{1}^{\left(1\right)},\cdots ,x_{t_{1}}^{\left(1\right)},x_{1}^{\left(2\right)},\cdots ,x_{t_{2}}^{\left(2\right)},x_{0}$ are all not equal to each other.


**(2) The preparation of a sequence of quantum states**


Alice prepares a sequence of quantum states $\left\{|\varphi_{1}\rangle ,|\varphi_{2}\rangle ,\cdots ,|\varphi_{m}\rangle \right\}$ according to the key $S=\left(s_{1},s_{2},\cdots ,s_{m}\right)$ with the following rules:



$$\mathrm{if}s_{i}=0,\mathrm{then}|\varphi_{i}\rangle =|{\mathrm{GHZ}}_{100}\rangle ;$$


$$\mathrm{if}s_{i}=1,\mathrm{then}|\varphi_{i}\rangle =|{\mathrm{GHZ}}_{000}\rangle .$$



Alice then prepares a random sequence of quantum states $\left\{\phi_{1},\;\phi_{2},\cdots ,\;\phi_{L}\right\}$ with each $|\varphi_{j}\rangle$ randomly between $|{\mathrm{GHZ}}_{000}\rangle$ and $|{\mathrm{GHZ}}_{100}\rangle$, where L=m(1+ζ)(j∈{1,2,⋯,L}), and ζ is a factor in determining the size of the test sample.


**(3) Distribution of quantum state sequences**


Alice lets the first particles in the sequence $\left\{|\varphi_{1}\rangle ,|\varphi_{2}\rangle ,\cdots ,|\varphi_{m}\rangle \right\}$ form the sequence $G_{1}$, the second particles form the sequence $G_{2}$, and the third particles form the sequence $G_{3}$. Alice keeps all the particles in the sequence $G_{1}$ and does the phase shift operation $U\left(2\pi -S+S_{0}\right)$ on all the particles in the sequence $G_{1}$, where
$$\begin{array}{c} U\left(2\pi -S+S_{0}\right)=\left(\begin{array}{cc} 1 & 0\\ 0 & e^{i\cdot (2\pi -S+S_{0})} \end{array}\right). \end{array}$$

**(4)** Alice forms the sequence $H_{1}$ with the first particles in the sequence $\{|\phi_{1}\rangle ,|\phi_{2}\rangle ,\cdots$, $|\phi_{m}\rangle \}$, the sequence $H_{2}$ with the second particle, and the sequence $H_{3}$ with the third particles. Alice takes random particles from the sequences $G_{2}$ and $H_{2}$ and sends them to the participant $P_{i}^{\left(1\right)}\left(i\in \left\{1,2,\cdots ,t_{1}\right\}\right)$ from the set $P^{\left(1\right)}$. Alice takes some particles from the sequences $G_{3}$ and $H_{3}$ randomly and then sends them to the participant $P_{1}^{\left(2\right)}$ from the set $P^{\left(2\right)}$,

Alice records the serial numbers of the particles when they are sent from $G_{i}$ and $H_{i}$ (i=2,3), and Alice herself keeps all the particles from $G_{1}$ and $H_{1}$.

The structure of the quantum network between Alice and all participants in this scheme is illustrated in Figure 1.


**(5) Secret reconstruction phase**


The process of reconstructing the secret by the participant $P_{i}^{\left(1\right)}$ from the set $P^{\left(1\right)}$ is given here.

The participant $P_{r}^{\left(j\right)}$ first calculates the shadow $S_{r}^{\left(j\right)}$ of the share.

when j=1,
$$\begin{array}{c} S_{r}^{\left(1\right)}=f\left(x_{r}^{\left(1\right)}\right)\prod_{\begin{array}{c} 1\leq i\leq t_{1}\\ i\neq r \end{array}}\frac{x_{i}^{\left(1\right)}}{\left(x_{i}^{\left(1\right)}-x_{r}^{\left(1\right)}\right)}\prod_{1\leq j\leq t_{2}}\frac{x_{j}^{\left(2\right)}}{\left(x_{j}^{\left(2\right)}-x_{r}^{\left(1\right)}\right)} \end{array}$$
where $r\in \{1,2,\cdots ,t_{1}\}$.

When j=2,
$$\begin{array}{c} S_{r}^{\left(2\right)}=f\left(x_{r}^{\left(2\right)}\right)\prod_{1\leq i\leq t_{1}}\frac{x_{i}^{\left(1\right)}}{\left(x_{i}^{\left(1\right)}-x_{r}^{\left(2\right)}\right)}\cdot \prod_{\begin{array}{c} 1\leq j\leq t_{2}\\ j\neq r \end{array}}\frac{x_{j}^{\left(2\right)}}{\left(x_{j}^{\left(2\right)}-x_{r}^{\left(2\right)}\right)} \end{array}$$
where $r\in \{1,2,\cdots ,t_{2}\}$.


**(6) Transferring particles to $P_{i}^{\left(1\right)}$ and $P_{1}^{\left(2\right)}$**


After participants $P_{i}^{\left(1\right)}$ and $P_{1}^{\left(2\right)}$ each receive the particle sequences $G_{2},H_{2}$ and $G_{3},H_{3}$, Alice tells $P_{i}^{\left(1\right)}$ and $P_{1}^{\left(2\right)}$ about the positions of these particles in the sequences $G_{2},H_{2}$ and $G_{3},H_{3}$, respectively. The participant $P_{1}^{\left(2\right)}$ does a phase shift of $U_{3}\left(S_{1}^{\left(2\right)}\right)$ for each particle from $G_{3}$. Then, participants $P_{i}^{\left(1\right)}$ and $P_{1}^{\left(2\right)}$ send the particle sequences $G_{2},H_{2}$ and $G_{3},H_{3}$ to participants $P_{\left(i+1\right)\mathrm{mod}t_{1}}^{\left(1\right)}$ and $P_{1}^{\left(2\right)}$ and tells them about the position of each particle in the sequences $G_{2},H_{2}$ and $G_{3},H_{3}$, respectively.


**(7) Transferring particles to $P_{\left(i+1\right)\mathrm{mod}t_{1}}^{\left(1\right)}$ and $P_{2}^{\left(2\right)}$**


The participants $P_{\left(i+1\right)\mathrm{mod}t_{1}}^{\left(1\right)}$ and $P_{2}^{\left(2\right)}$ do a phase shift of $U_{2}\left(S_{\left(i+1\right)\mathrm{mod}t_{1}}^{\left(1\right)}\right)$ and $U_{3}\left(S_{2}^{\left(2\right)}\right)$ for each particle in $G_{2}$ and $G_{3}$, respectively. Then, they send the particle sequences $G_{2},H_{2}$ and $G_{3},H_{3}$ to participant $P_{\left(i+2\right)\mathrm{mod}t_{1}}^{\left(1\right)}$ and participant $P_{3}^{\left(2\right)}$, respectively, and tell them about the position of each particle in the sequences $G_{2},H_{2}$ and $G_{3},H_{3}$.


**(8) Transferring particles to $P_{\left(i+t_{1}-1\right)\mathrm{mod}t_{1}}^{\left(1\right)}$ and $P_{t_{2}}^{\left(2\right)}$**


Follow the above steps and so on, until $P_{\left(i+t_{1}-1\right)\mathrm{mod}t_{1}}^{\left(1\right)}$ and $P_{t_{2}}^{\left(2\right)}$ receive the particle sequences $G_{2},H_{2}$ and $G_{3},H_{3}$ from $P_{\left(i+t_{1}-2\right)\mathrm{mod}t_{1}}^{\left(1\right)}$ and $P_{t_{2}-1}^{\left(2\right)}$, respectively, $P_{\left(i+t_{1}-1\right)\mathrm{mod}t_{1}}^{\left(1\right)}$ and $P_{t_{2}}^{\left(2\right)}$ do phase shift each of the particles in $G_{2}$ and $G_{3}$ by $U_{2}\left(S_{\left(i+t_{1}-1\right)\mathrm{mod}t_{1}}^{\left(1\right)}\right)$ and $U_{3}\left(S_{t_{2}}^{\left(2\right)}\right)$, respectively. Then, $P_{\left(i+t_{1}-1\right)\mathrm{mod}t_{1}}^{\left(1\right)}$ sends the particle sequence $G_{2},H_{2}$ back to $P_{i}^{\left(1\right)}$. At the same time, $P_{t_{2}}^{\left(2\right)}$ also sends the particle sequence $G_{3},H_{3}$ back to $P_{i}^{\left(1\right)}$ and tells $P_{i}^{\left(1\right)}$ the position of each particle from the particle sequence $G_{2},H_{2}$ and $G_{3},H_{3}$. Finally, participant $P_{i}^{\left(1\right)}$ does a phase shift of $U_{2}\left(S_{i}^{\left(1\right)}\right)$ for each of the particles from the sequence $G_{2}$.

### 3.3. Detecting Eavesdropping

Alice uses the measurement base $B_{x}=\{|x\rangle ,|-x\rangle \}$ to measure the particles in the sequence $H_{1}$. Then, $P_{i}^{\left(1\right)}$ measures the corresponding particles in the sequences $H_{2}$ and $H_{3}$ using either the measurement base $B_{x}=\{|x\rangle ,|-x\rangle \}$ or $B_{y}=\{|y\rangle ,|-y\rangle \}$. Where the measurement bases |+x〉,|−x〉 and |+y〉,|−y〉 are represented by the base vector |0〉,|1〉 as
$$\begin{array}{c} \begin{array}{c} |+x\rangle =\frac{1}{\sqrt{2}}(|0\rangle +|1\rangle ),|-x\rangle =\frac{1}{\sqrt{2}}(|0\rangle -|1\rangle ),\\ |+y\rangle =\frac{1}{\sqrt{2}}(|0\rangle +i|1\rangle ),|-y\rangle =\frac{1}{\sqrt{2}}(|0\rangle -i|1\rangle ). \end{array} \end{array}$$

For $|{\mathrm{GHZ}}_{000}\rangle$ and $|{\mathrm{GHZ}}_{100}\rangle$, when Alice and $P_{i}^{\left(1\right)}$ measure with the above bases, the following four combinations of measurement bases with associated properties occur.

(1) If both sides measure $|{\mathrm{GHZ}}_{000}\rangle$ with $B_{x},B_{x},B_{x}$-bases, then
$$\begin{array}{c} \begin{array}{cc} |{\mathrm{GHZ}}_{000}\rangle & =\frac{1}{2}(|+x\rangle |+x\rangle |+x\rangle +|+x\rangle |-x\rangle |-x\rangle \\ & +|-x\rangle |-x\rangle |+x\rangle +|-x\rangle |+x\rangle |-x\rangle ). \end{array} \end{array}$$

(2) If both sides measure $|{\mathrm{GHZ}}_{000}\rangle$ with $B_{x},B_{y},B_{y}$-bases, then
$$\begin{array}{c} \begin{array}{cc} |{\mathrm{GHZ}}_{000}\rangle & =\frac{1}{2}(|+x\rangle |+y\rangle |-y\rangle +|-x\rangle |-y\rangle |+y\rangle \\ & +|-x\rangle |+y\rangle |+y\rangle +|+x\rangle |-y\rangle |-y\rangle ). \end{array} \end{array}$$

(3) If both sides measure $|{\mathrm{GHZ}}_{100}\rangle$ with $B_{x},B_{x},B_{x}$-bases, then
$$\begin{array}{c} \begin{array}{cc} |{\mathrm{GHZ}}_{100}\rangle & =\frac{1}{2}(|+x\rangle |+x\rangle |-x\rangle +|+x\rangle |-x\rangle |+x\rangle \\ & +|-x\rangle |-x\rangle |-x\rangle +|-x\rangle |+x\rangle |+x\rangle ). \end{array} \end{array}$$

(4) If both sides measure $|{\mathrm{GHZ}}_{100}\rangle$ with $B_{x},B_{y},B_{y}$-bases, then
$$\begin{array}{c} \begin{array}{cc} |{\mathrm{GHZ}}_{100}\rangle & =\frac{1}{2}(|+x\rangle |+y\rangle |+y\rangle +|+x\rangle |-y\rangle |-y\rangle \\ & +|-x\rangle |+y\rangle |-y\rangle +|-x\rangle |-y\rangle |+y\rangle ). \end{array} \end{array}$$

From the above results, it is clear that, when Alice measures the particles from the sequence $H_{1}$ with basis $B_{x}$, $P_{i}^{\left(1\right)}$ measures the corresponding particles, which he holds using the basis $B_{x}$ or $B_{y}$, then the measurements are correlated; see Table 1.

When these measurements are completed, Alice asks $P_{i}^{\left(1\right)}$ to tell her the results of their measurements; however, Alice will not open her measurement base. Then, Alice statistically determines the error rate from Table 1. If the error rate is above a certain threshold, this communication is abandoned. Otherwise, this protocol continues.

### 3.4. Measuring Information Particles

When Alice and $P_{i}^{\left(1\right)}$ confirms that the channel is secure, Alice measures her particle sequence $G_{1}$, and $P_{i}^{\left(1\right)}$ measures her particle sequence $G_{2}$ and $G_{3}$.

(1) First, $P_{i}^{\left(1\right)}$ secretly selects the random sequence $K_{1}^{\left(1,i\right)}=\left(k_{1}^{\left(1,1,i\right)},k_{2}^{\left(1,1,i\right)},\cdots ,k_{m}^{\left(1,1,i\right)}\right)$ consisting of 0 and 1 and uses the following rules to measure the particles from sequence $G_{2}$ and $G_{3}$ from their own hand, and the rules for the measuring base are as follows:

When the *j*-th bit of $K_{1}^{\left(1,i\right)}$ is equal to 0, it selects the $B_{x}$ base.

When the *j*-th bit of $K_{1}^{\left(1,i\right)}$ is equal to 1, it selects the $B_{y}$ base.

(2) $P_{i}^{\left(1\right)}$ uses the same base to measure the particles from sequences $G_{2}$ and $G_{3}$ and records these results. These measurements are converted into binary numbers—that is, |+x〉 and |+y〉 correspond to 1, while |−x〉 and |−y〉 correspond to 0, which in turn constitute two subkeys of $P_{i}^{\left(1\right)}$ and are recorded as $K_{2}^{\left(1,i\right)}$ and $K_{3}^{\left(1,i\right)}$, respectively.

(3) Alice measures the corresponding particle using the base $B_{x}$ from the sequence $G_{1}$ and encodes these results as a bit string $K_{A}^{\left(1,i\right)}$. The encoding rule is that it is recorded as 1 when the measurement result is |+x〉 and 0 when the measurement result is |−x〉. Alice then sends $E_{f\left(x_{i}^{\left(1\right)}\right)}\left(K_{A}^{\left(1,i\right)}\right)$ secretly to $P_{i}^{\left(1\right)}$. $P_{i}^{\left(1\right)}$ receives $E_{f\left(x_{i}^{\left(1\right)}\right)}\left(K_{A}^{\left(1,i\right)}\right)$ and decrypts it using $f(x_{i}^{\left(1\right)})$ to obtain $K_{A}^{\left(1,i\right)}$.

### 3.5. Reconstruction and Detection of Keys

$P_{i}^{\left(1\right)}$ computes $K_{1}^{\left(1,i\right)}⊕K_{2}^{\left(1,i\right)}⊕K_{3}^{\left(1,i\right)}⊕K_{A}^{\left(1,i\right)}$, and verifies whether $H(K_{1}^{\left(1,i\right)}⊕K_{2}^{\left(1,i\right)}⊕K_{3}^{\left(1,i\right)}$$⊕K_{A}^{\left(1,i\right)})=H\left(S\right)$ holds. If this equation holds, $P_{i}^{\left(1\right)}$ retains *S* as the shared key. Otherwise, he judges that some of the participants had offered a false share in the secret recovery process and can therefore abandon this round.

Next, we present the process in which participant $P_{i}^{\left(1\right)}\left(i\in \left\{1,2,\cdots ,t_{1}\right\}\right)$ from the set $P^{\left(1\right)}$ gives their shares to all participants. For the ease of presentation, we arranged the order in which the participants from the set $P^{\left(2\right)}$ pass the particles with the natural order of their numbers.

Figure 2a shows the transferring process of the information particle in the *q*-th GHZ state where the GHZ state consists of a green ball $G_{1}^{\left(q\right)}$, red ball $G_{2}^{\left(q\right)}$, and blue ball $G_{3}^{\left(q\right)}$, q∈{1,2,⋯,m}. First, Alice does the $U(2\pi -S+S_{0})$ phase operation to particle $G_{1}^{\left(q\right)}$. Then, Figure 2a gives the process in which participant $P_{i}^{\left(1\right)}$ from the set $P^{\left(1\right)}$ shares the sub-shares of all participants, and Figure 2b gives the process in which participant $P_{i}^{\left(2\right)}$ from the set $P^{\left(2\right)}$ shares the sub-shares of all participants.

The process participant $P_{i}^{\left(2\right)}\left(i\in \left\{1,2,\cdots ,t_{2}\right\}\right)$ from the set $P^{\left(2\right)}$ shares the sub-shares for all participants, which is similar to the above process.

## 4. Performance Analysis

### 4.1. Correctness

**Theorem** **3.**
*When Alice and t−1 participants from two sets $P^{\left(1\right)}$ and $P^{\left(2\right)}$ perform a phase shift operation on the particles in the GHZ quantum state sequence $\{|\varphi_{1}\rangle ,|\varphi_{2}\rangle ,\cdots ,$$|\varphi_{m}\rangle \}$, then Alice and $P_{i}^{\left(1\right)}$$\left(i\in \left\{1,2,\cdots ,t_{1}\right\}\right)$ complete the corresponding measurement. $P_{i}^{\left(1\right)}$ will finally obtain the distributed key sequence S.*


**Proof.** First, if Alice and $P_{i}^{\left(1\right)}$ confirm that the channel is secure, the quantum state $|\varphi_{j}\rangle$ will become $U_{1}\left(2\pi -S+S_{0}\right)⊗I⊗I|\varphi_{j}\rangle$ when Alice has performed the phase shift operation j∈{1,2,⋯,m}. In the recovery phase, according to Theorems 1 and 2, after t−1 participants have performed a phase shift operation, the quantum state $U_{1}\left(2\pi -S+S_{0}\right)⊗I⊗I|\varphi_{j}\rangle$ will become
$$\begin{array}{c} I⊗U_{2}\left[\left(2\pi -S+S_{0}\right)+\left(\sum_{r=1}^{t_{1}}S_{r}^{\left(1\right)}+\sum_{r=1}^{t_{2}}S_{r}^{\left(2\right)}\right)\right]⊗I|\varphi_{i}\rangle =I⊗U_{2}\left(2\pi \right)⊗I|\varphi_{j}\rangle =|\varphi_{j}\rangle . \end{array}$$Here, it is easy to see from Lagrange’s formula that $S=S_{0}+\sum_{r=1}^{t_{1}}S_{r}^{\left(1\right)}+\sum_{r=1}^{t_{2}}S_{r}^{\left(2\right)}$. Thus, $P_{i}^{\left(1\right)}$ will recover the sequence of quantum states $\{|\varphi_{1}\rangle ,|\varphi_{2}\rangle ,\cdots ,|\varphi_{m}\rangle \}$.Next, we will prove that, when Alice and $P_{i}^{\left(1\right)}$ confirmed that the channel is security, $P_{i}^{\left(1\right)}$ will obtain the following equation according to this protocol, i.e.,
(12)$$S=K_{1}^{\left(1,i\right)}⊕K_{2}^{\left(1,i\right)}⊕K_{3}^{\left(1,i\right)}⊕K_{A}^{\left(1,i\right)}.$$Here, $S=(s_{1},s_{2},\cdots ,s_{m})$, $s_{i}\in \left\{0,1\right\},i=1,2,\cdots ,m$.Let
$$\begin{array}{c} M=\left(\begin{array}{c} S\\ K_{A}^{\left(1,i\right)}\\ K_{1}^{\left(1,i\right)}\\ K_{2}^{\left(1,i\right)}\\ K_{3}^{\left(1,i\right)} \end{array}\right), \end{array}$$
where *M* is a 5×m matrix. Let us first analyze the value of the *j*-th column of this matrix *M*. □

**Case (1)** When the *j*-th portion of *S* is 0, i.e., the *j*-th entangled state of *S* is encoded as $|{\mathrm{GHZ}}_{100}\rangle$. In this case, there are two ways that $K_{1}^{\left(1,i\right)}$ can be evaluated.

**(1.1)** The *j*-th element of $K_{1}^{\left(1,i\right)}$ takes the value 1. This means that $P_{i}^{\left(1\right)}$ measures the particles in the corresponding $G_{2}$ and $G_{3}$ with the $B_{y}$-base, and then the *j*-th column of $\left(K_{2}^{\left(1,i\right)},K_{3}^{\left(1,i\right)}\right)$ will take the following two cases.
$$\begin{array}{c} \left(i\right)\left(\begin{array}{c} 1\\ 1 \end{array}\right),\left(\begin{array}{c} 0\\ 0 \end{array}\right);\left(\mathrm{ii}\right)\left(\begin{array}{c} 1\\ 0 \end{array}\right),\left(\begin{array}{c} 0\\ 1 \end{array}\right). \end{array}$$

In case (i), the *j*-th element of K1(1,i) is 1; and in case (ii), the *j*-th element of K1(1,i) is 0.

From the above analysis, it follows that the *j*-th column of *M* is the following four cases.
(13)$$\begin{array}{c} {\left(\begin{array}{ccccc} 0 & 1 & 1 & 1 & 1 \end{array}\right)}^{T},{\left(\begin{array}{ccccc} 0 & 1 & 1 & 0 & 0 \end{array}\right)}^{T},\\ {\left(\begin{array}{ccccc} 0 & 0 & 1 & 1 & 0 \end{array}\right)}^{T},{\left(\begin{array}{ccccc} 0 & 0 & 1 & 0 & 1 \end{array}\right)}^{T}. \end{array}$$

**(1.2)** The *j*-th element of $K_{1}^{\left(1,i\right)}$ takes the value 0. This means that $P_{i}^{\left(1\right)}$ measures the corresponding particles in $G_{2}$ and $G_{3}$ with the $B_{x}$ base, and the *j*-th column element of $\left(K_{2}^{\left(1,i\right)},K_{3}^{\left(1,i\right)}\right)$ will take the following two cases.
$$\begin{array}{c} \begin{array}{c} \left(i\right)\left(\begin{array}{c} 1\\ 0 \end{array}\right),\left(\begin{array}{c} 0\\ 1 \end{array}\right);\left(\mathrm{ii}\right)\left(\begin{array}{c} 1\\ 1 \end{array}\right),\left(\begin{array}{c} 0\\ 0 \end{array}\right). \end{array} \end{array}$$

In case (i), the *j*-th element of $K_{1}^{\left(1,i\right)}$ is 1; and in case (ii), the *j*-th element of $K_{1}^{\left(1,i\right)}$ is 0.

From the above analysis, it is clear that the *j*-th column element of *M* is in the following four cases.
(14)$$\begin{array}{c} {\left(\begin{array}{ccccc} 0 & 1 & 0 & 1 & 0 \end{array}\right)}^{T},{\left(\begin{array}{ccccc} 0 & 1 & 0 & 0 & 1 \end{array}\right)}^{T},\\ {\left(\begin{array}{ccccc} 0 & 0 & 0 & 1 & 1 \end{array}\right)}^{T},{\left(\begin{array}{ccccc} 0 & 0 & 0 & 0 & 0 \end{array}\right)}^{T}. \end{array}$$

**Case (2)** When the *j*-th portion of *S* is 1, i.e., the *j*-th entangled state that *S* is encoded as $|{\mathrm{GHZ}}_{000}\rangle$, in this case, there are two ways that $K_{1}^{\left(1,i\right)}$ can be evaluated.

**(2.1)** The *j*-th element of $K_{1}^{\left(1,i\right)}$ takes the value 1. This means that $P_{i}^{\left(1\right)}$ measures the particles in the corresponding $G_{2}$ and $G_{3}$ with the $B_{y}$-base, and then the *j*-th column of $\left(K_{2}^{\left(1,i\right)},K_{3}^{\left(1,i\right)}\right)$ takes the following two cases.
$$\begin{array}{c} \begin{array}{c} \left(i\right)\left(\begin{array}{c} 1\\ 0 \end{array}\right),\left(\begin{array}{c} 0\\ 1 \end{array}\right);\left(\mathrm{ii}\right)\left(\begin{array}{c} 1\\ 1 \end{array}\right),\left(\begin{array}{c} 0\\ 0 \end{array}\right). \end{array} \end{array}$$

In case (i), the *j*-th element of K1(1,i) is 1; and in case (ii), the *j*-th element of K1(1,i) is 0.

From the above analysis, it follows that the *j*-th column of *M* is the following four cases.
(15)$$\begin{array}{c} {\left(\begin{array}{ccccc} 1 & 1 & 1 & 1 & 0 \end{array}\right)}^{T},{\left(\begin{array}{ccccc} 1 & 1 & 1 & 0 & 1 \end{array}\right)}^{T},\\ {\left(\begin{array}{ccccc} 1 & 0 & 1 & 1 & 1 \end{array}\right)}^{T},{\left(\begin{array}{ccccc} 1 & 0 & 1 & 0 & 0 \end{array}\right)}^{T}. \end{array}$$

**(2.2)** The *j*-th element of $K_{1}^{\left(1,i\right)}$ takes the value 0. This means that $P_{i}^{\left(1\right)}$ measures the particles in the corresponding $G_{2}$ and $G_{3}$ with the $B_{x}$ base, and the *j*-th column element of $\left(K_{2}^{\left(1,i\right)},K_{3}^{\left(1,i\right)}\right)$ is either
$$\begin{array}{c} \begin{array}{c} \left(i\right)\left(\begin{array}{c} 1\\ 1 \end{array}\right),\left(\begin{array}{c} 0\\ 0 \end{array}\right);\left(\mathrm{ii}\right)\left(\begin{array}{c} 1\\ 0 \end{array}\right),\left(\begin{array}{c} 0\\ 1 \end{array}\right). \end{array} \end{array}$$

In case (i), the *j*-th element of $K_{1}^{\left(1,i\right)}$ is 1; and in case (ii), the *j*-th element of $K_{1}^{\left(1,i\right)}$ is 0.

From the above analysis, it is clear that the *j*-th column of *M* is in the following four cases.
(16)$$\begin{array}{c} {\left(\begin{array}{ccccc} 1 & 1 & 0 & 1 & 1 \end{array}\right)}^{T},{\left(\begin{array}{ccccc} 1 & 1 & 0 & 0 & 0 \end{array}\right)}^{T},\\ {\left(\begin{array}{ccccc} 1 & 0 & 0 & 1 & 0 \end{array}\right)}^{T},{\left(\begin{array}{ccccc} 1 & 0 & 0 & 0 & 1 \end{array}\right)}^{T}. \end{array}$$

Equations (13)–(16) give all the values of the *j*-th row element of the matrix *M*, which shows that the first column of *M* is exactly the sum of the remaining four rows; thus, we have proven that $S=K_{A}^{\left(1,i\right)}⊕K_{1}^{\left(1,i\right)}⊕K_{2}^{\left(1,i\right)}⊕K_{3}^{\left(1,i\right)}$. Therefore, $P_{i}^{\left(1\right)}\left(i\in \left\{1,2,\cdots ,t_{1}\right\}\right)$ will finally obtain the key *S* distributed by Alice.

Using these steps of $P_{i}^{\left(1\right)}$, reconstructing the key in Theorem 3, participant $P_{i}^{\left(2\right)}\left(i\in \left\{1,2,\cdots ,i-1,i+1,\cdots ,t_{2}\right\}\right)$ can also obtain the distributed key *S* in the same way.

### 4.2. Security Analysis of the Protocol

The security of the protocol relies on the decoy particle sequences randomly inserted during the transmission of quantum information. In this protocol, Alice sends randomly selected particles from sequences $G_{2}$ and $H_{2}$ to participant $P_{i}^{\left(1\right)}$ from the set $P^{\left(1\right)}$ and randomly selected particles from sequences $G_{3}$ and $H_{3}$ to participant $P_{1}^{\left(2\right)}$ from the set $P^{\left(2\right)}$. Then, when these information particles are transmitted according to Figure 2a, the decoy particles are also interspersed with the information particle sequence until $P_{i}^{\left(1\right)}$ receives the particles from sequence $G_{2}$ and $G_{3}$, and the particles in sequence $H_{2}$ and $H_{3}$. $P_{i}^{\left(1\right)}$ will detect this round of communication by detecting particles from the decoy state sequences $H_{2}$ and $H_{3}$.

If the measurement is above a certain threshold, it indicates that there is the presence of an external eavesdropper, Eve. This means that any attack from an external eavesdropper will be detected with a certain probability during the eavesdropping inspection phase. That is to say, this protocol can prevent eavesdropping from external attackers, thus, achieving the security of the scheme. In essence, this prevents eavesdropping by external attackers. The types of attacks that the protocol can resist are further discussed below based on certain properties.

#### 4.2.1. Direct Measurement by the Attacker

If the eavesdropper Eve attacks the two particles in the GHZ state by measuring the two transmitting particles, which are distributed to participants from the set $P^{\left(1\right)}$ and $P^{\left(2\right)}$, respectively. However, Eve cannot measure both particles at the same time, she can only measure one of them.

Assuming that the *i*-th initial GHZ state is $|\varphi_{i}\rangle =\left[\frac{1}{\sqrt{2}}(|000\rangle +\left|111\rangle \right)\right]$, then, after Alice performs the phase shift operation $U_{1}\left(2\pi -S+S_{0}\right)$ on the first particle in the quantum state $|\varphi_{i}\rangle$, the participants perform the phase shift operation on $|\varphi_{i}\rangle$ in turn.

After Alice has operated on the quantum state $|\varphi_{i}\rangle$, suppose that $l_{1}\left(l_{1}\in \left\{1,2,\cdots ,t_{1}\right\}\right)$ participants from the set $P^{\left(1\right)}$ have performed $l_{1}$ operations and $l_{2}\left(l_{2}\in \left\{1,2,\cdots ,t_{2}\right\}\right)$ participants from the set $P^{\left(2\right)}$ have performed $l_{2}$ operations. Using Theorems 1 and 2, it follows that the quantum state $|\varphi_{i}\rangle$ then becomes
(17)$$|\varphi_{i}^{\prime }\rangle =\frac{1}{\sqrt{2}}(|000\rangle +e^{i\cdot \alpha }|111\rangle ),$$
where $\alpha =\left(2\pi -S+S_{0}\right)+\sum_{r=1}^{l_{1}}S_{\left(i+r\right)\;\mathrm{mod}\;t_{1}}^{\left(1\right)}+\sum_{r=1}^{l_{2}}S_{r\;\mathrm{mod}\;t_{2}}^{\left(2\right)}$.

From Equation (Equation 6), the probability of each particle in the GHZ state existing in state |0〉 or |1〉 is
$$\begin{array}{c} {\left|\frac{1}{\sqrt{2}}\right|}^{2}+{\left|\frac{1}{\sqrt{2}}e^{i\cdot \alpha }\right|}^{2}=\frac{1}{2}. \end{array}$$

Furthermore, since $l_{i}\left(i=1,2\right)$ was arbitrary, it was impossible for Eve to obtain any useful information by measuring the GHZ particles that had been passed on.

#### 4.2.2. Interception–Relaunch Attack

Eve may have intercepted the particles in the participants’ hands and sent her own counterfeit particles to the participants. In this case, Eve cannot obtain any information about the key. This is because it is known from this protocol construction process that the entire key is obtained through the post-processing phase after the transferring the particle sequences $G^{\left(2\right)}$ and $G^{\left(3\right)}$ from the GHZ sequence $\left\{|\varphi_{1}\rangle ,|\varphi_{2}\rangle ,\ldots ,|\varphi_{m}\rangle \right\}$, during which the original quantum sequence $\left\{|\varphi_{1}\rangle ,|\varphi_{2}\rangle ,\ldots ,|\varphi_{m}\rangle \right\}$ requires the phase shifting operations of each participant, and these unitary matrices are known only by each participant.

Even if Eve tries to intercept the last round of particles, we suppose that the $P_{i}^{\left(1\right)}\left(i\in \left\{1,2,\cdots ,t_{1}\right\}\right)$ can reconstruct the key. Specifically, if Eve had managed to intercept particles from $P_{\left(i+t_{1}-1\right)modt_{1}}^{\left(1\right)}$ to $P_{i}^{\left(1\right)}$ or $P_{t_{2}}^{\left(2\right)}$ to $P_{i}^{\left(1\right)}$, it would also have been impossible for Eve to have obtained particles from the original quantum state sequence $\left\{|\varphi_{1}\rangle ,|\varphi_{2}\rangle ,\ldots ,|\varphi_{m}\rangle \right\}$, since the original quantum state sequence $\left\{|\varphi_{1}\rangle ,|\varphi_{2}\rangle ,\ldots ,|\varphi_{m}\rangle \right\}$ could only be restored after $P_{i}^{\left(1\right)}$ had received the returned particles and performed a phase shift operation. As a result, Eve could not obtain any information about the key.

#### 4.2.3. Entanglement Measurement Attack

Eve tries to launch an entanglement attack when the participants from the sets $P^{\left(1\right)}$ and $P^{\left(2\right)}$ each transport particles. Let us set that, when the participant $P_{i}^{\left(1\right)}\left(i\in \left\{1,2,\cdots ,t_{1}\right\}\right)$ from the set $P^{\left(1\right)}$ passes $G_{2}^{\left(u\right)}$ particles of $|\varphi_{u}\rangle$-state to $P_{i+1(\mathrm{mod}t_{1})}^{\left(1\right)}$, Eve launches an entanglement attack, and the auxiliary qubit is $|E_{init}\rangle$, while the entanglement bit and the auxiliary qubit form a hybrid quantum state,
$$\begin{array}{c} |\Psi_{AP_{i}^{\left(1\right)}P_{j}^{\left(2\right)}E}\rangle =|\varphi_{u}\rangle ⊗|E_{\mathrm{init}}\rangle , \end{array}$$
where $A,P_{i}^{\left(1\right)},P_{j}^{\left(2\right)},E$ denote the holders of four particles from the entangled state $|\Psi_{AP_{i}^{\left(1\right)}P_{j}^{\left(2\right)}E}\rangle$ Alice, $P_{i}^{\left(1\right)},P_{j}^{\left(2\right)}$ and Eve, respectively, where $i\in \left\{1,2,\cdots ,t_{1}\right\},j\in \left\{1,2,\cdots ,t_{2}\right\}$.

The attacker applies a quantum operation to $|\Psi_{AP_{i}^{\left(1\right)}P_{j}^{\left(2\right)}E}\rangle$ by a unitary transformation U(ε) to obtain
$$\begin{array}{c} \begin{array}{cc} U\left(\varepsilon \right)|\Psi_{AP_{i}^{\left(1\right)}P_{j}^{\left(2\right)}E}\rangle & =U\left(\varepsilon \right)\left(|\varphi_{u}\rangle ⊗|E_{\mathrm{init}\;}\rangle \right)\\ & =U\left(\varepsilon \right)\left[\frac{1}{\sqrt{2}}\left(|0_{A}0_{P_{i}}0_{P_{j}}\rangle +|1_{A}1_{p_{i}^{\left(1\right)}}1_{P_{j}^{\left(2\right)}}\rangle \right)⊗|E_{\mathrm{init}\;}\rangle \right]. \end{array} \end{array}$$

Since $|E_{\mathrm{init}\;}\rangle$ is a qubit $|0_{E}\rangle$ or $|1_{E}\rangle$, let us say that $|E_{\mathrm{init}\;}\rangle =|0_{E}\rangle$, and let the U(ε) act on the particles held by $P_{i}^{\left(1\right)}$ and Eve. Then, we have
$$\begin{array}{c} \begin{array}{c} \begin{array}{cc} U\left(\varepsilon \right)|\Psi_{AP_{i}^{\left(1\right)}P_{j}^{\left(2\right)}E}\rangle & =U\left(\varepsilon \right)\left(|\Phi^{+}\rangle ⊗|0_{E}\rangle \right)\\ & =U\left(\varepsilon \right)\left[\frac{1}{\sqrt{2}}\left(|0_{A}0_{P_{i}^{\left(1\right)}}0_{P_{j}^{\left(2\right)}}\rangle +|1_{A}1_{P_{i}^{\left(1\right)}}1_{P_{j}^{\left(2\right)}}\rangle \right)⊗|0_{E}\rangle \right]\\ & =U\left(\varepsilon \right)\left[\frac{1}{\sqrt{2}}\left(|0_{A}0_{P_{1}^{\left(1\right)}}0_{P_{j}^{\left(2\right)}}0_{E}\rangle +|1_{A}1_{P_{i}^{\left(1\right)}}1_{P_{j}^{\left(2\right)}}0_{E}\rangle \right)\right]. \end{array} \end{array} \end{array}$$

According to the Schmidt decomposition of the quantum state, let
$$\begin{array}{c} \begin{array}{c} \begin{array}{c} U\left(\varepsilon \right)|0_{P_{i}^{\left(1\right)}}0_{E}\rangle =|0_{P_{i}^{\left(1\right)}}\rangle ⊗E_{1}+|1_{P_{1}^{\left(1\right)}}\rangle ⊗E_{2},\\ U\left(\varepsilon \right)|1_{P_{i}^{\left(l\right)}}0_{E}\rangle =|0_{P_{i}^{\left(1\right)}}\rangle ⊗{\tilde{E}}_{1}+|1_{P_{i}^{\left(1\right)}}\rangle ⊗{\tilde{E}}_{2}. \end{array} \end{array} \end{array}$$
and then
(18)$$\begin{array}{c} \begin{array}{c} U\left(\varepsilon \right)|\Psi_{AP_{i}^{\left(1\right)}P_{j}^{\left(2\right)}E}\rangle =U\left(\varepsilon \right)\left[\frac{1}{\sqrt{2}}\left(|0_{A}0_{P_{i}^{\left(1\right)}}0_{P_{j}^{\left(2\right)}}0_{E}\rangle +|1_{A}1_{P_{i}^{\left(1\right)}}1_{p_{j}^{\left(2\right)}}1_{E}\rangle \right)\right]\\ =|0_{A}0_{P_{t}^{\left(1\right)}}0_{P_{j}^{\left(2\right)}}\rangle ⊗|E_{1}\rangle +|0_{A}0_{P_{t}^{\left(1\right)}}1_{P_{j}^{\left(2\right)}}\rangle ⊗|E_{2}\rangle \\ +|1_{A}1_{p_{t}^{\left(1\right)}}0_{P_{j}^{\left(2\right)}}\rangle ⊗|{\tilde{E}}_{1}\rangle +|1_{A}1_{p_{t}^{\left(1\right)}}1_{P_{j}^{\left(2\right)}}\rangle ⊗|{\tilde{E}}_{2}\rangle . \end{array} \end{array}$$
where $|E_{1}\rangle ⊥|E_{2}\rangle ,|{\tilde{E}}_{1}\rangle ⊥|{\tilde{E}}_{2}\rangle ,\;\mathrm{and}\;\langle E_{1}|{\tilde{E}}_{2}\rangle +\langle E_{2}|{\tilde{E}}_{1}\rangle =0$.

From Equation (Equation 12) and the key generation process of this protocol, it is clear that Eve cannot obtain any information about the key from $U\left(\varepsilon \right)|\Psi_{AP_{i}^{\left(1\right)}P_{j}^{\left(2\right)}E}\rangle$.

## 5. Comparisons

We analyzed and compared the proposed QSS protocol with several similar existing QSS protocols—namely, RP2015 [22], YGWQZW2015 [26], BLWLL2018 [16], and LYZ2021 [25], based on four parameters, including the universality of the scheme, communication costs, computational costs, and the efficiency of the scheme as shown in Table 2. First, the similarity of these schemes is that their access structure is a kind of special threshold structure.

Universality is shown in Table 2, which includes the theoretical basis of these schemes’ dependency, the adaptive access structure, the trajectory of information particles, the number of participants who ultimately calculate the key, and the key validation. Communication costs are based on the transmitted particles, i.e., information particles and decoy particles. The cost is calculated according to the following three parameters: the unitary operation, the measurement operation, and the hash function. Finally, we give the efficiency of each scheme.

Information Efficiency η [32] is defined as $\eta =\frac{c}{q}$, in which *c* represents the number of classical bits shared and *q* represents the total number of qubits transmitted within a quantum channel. According to this efficiency formula, the information efficiency of a protocol sharing *m* classical information can be expressed as $\eta =\frac{m}{n_{1}m+n_{2}v}$, where $n_{1}$ represents the number of particles contained in each quantum state when *m* bits of classical information are converted to *m* quantum state information. $n_{2}$ represents the number of particles contained in each quantum state in which the eavesdropping is detected. Furthermore, *v* represents the number of quantum states in which the eavesdropping is detected. Let v=L when the detected particles are entangled, and let v=l when the detected particles are single photons.

For the sake of parameter uniformity, the communication and computational costs required to recover the key once for *t* participants in each scenario are calculated in Table 1. The following is an analysis and comparison among RP2015 [22], YGWQZW2015 [26], BLWLL2018 [16], LYZ2021 [25], and our proposal.

**(1) RP2015 Protocol** The RP2015 protocol distribution model is a tree structure, i.e., the distributor distributes *t* information particles from the GHZ state to *t* participants, all from the two-dimensional Hilbert space. *m* GHZ states are used to share *m* bits of classical information, while *L* GHZ states are applied to detect eavesdropping. Thus, the information efficiency of both schemes is $\frac{m}{t(m+L)}$. The access structure of the participants in this distribution model is a restricted threshold structure, which is also a fully bipartite graph structure.

**(2) YGWQZW2015 Protocol** The distribution YGWQZW2015 model is a tree structure, i.e., the distributor distributes *t* information particles from the GHZ state to *t* participants, unlike in the BLWLL2018 protocol [16] where these particles are all from the K-dimensional Hilbert space. *m* GHZ states are used to share *m* bits of classical information, while *L* GHZ states are applied to detect eavesdropping. Therefore, the information efficiency of this scheme is $\frac{m}{t(m+L)}$. The access structure of the participants in this allocation model belongs to the fully bipartite graph.

**(3) BLWLL2018 Protocol** This distribution model of the BLWLL2018 scheme is also a tree structure, i.e., the distributor distributes *t* information particles from the GHZ state to *t* participants, all of which come from the *k*-dimensional Hilbert space. *m* GHZ states are used to share *m* bits of classical information, while *L* GHZ states are applied to detect eavesdropping. Therefore, the information efficiency of this scheme is $\frac{m}{t(m+L)}$. Unlike the YGWQZW2015 protocol, the access structure of the participants in this distribution model is a restricted threshold structure and also a fully bipartite graph structure.

**(4) LYZ2021 Protocol** The LYZ2021 distribution model of the LYZ2021 scheme is a one circle structure, where the distributor distributes a particle of information from a generalized Bell state to one of the participants, and the particle is then passed through *t* participants in turn, where the two particles in the Bell state are from the *k*-dimensional Hilbert space. *m*-generalised Bell states are used to share *m*-bit classical information, while *lX*-bases and *Z*-bases are applied to detect eavesdropping. Thus, the information efficiency of the scheme is $\frac{m}{t(m+l)}$.

**(5) Our Protocol** This distribution model of our scheme is a bicyclic structure, i.e., the distributor distributes two information particles from the GHZ state to *t* participants according to the requirements of a fully bipartite graph structure, where the three particles from the GHZ state are from a three-dimensional Hilbert space. *m* GHZ states are used to share *m*-bit classical information, while *L* GHZ states are applied to detect eavesdropping. Thus, the information efficiency of both schemes is $\frac{m}{3(m+L)}$.

In the protocol proposed, the quantum states corresponding to the information particles and the detection particles are three-dimensional GHZ states, and are only detected between Alice and $P_{i}^{\left(1\right)}$ (or $P_{j}^{\left(2\right)}$), where (m+L) three-dimensional GHZ states are used as information quantum states and detection quantum states, *m*-dimensional bits of classical information are obtained, and the efficiency of the scheme is $\frac{m}{3(m+L)}$. It can be seen that the scheme in this paper significantly saves quantum resources and is significantly more efficient than the above schemes.

## 6. Conclusions

In this paper, we proposed an efficient quantum secret sharing scheme for restricted gated access structures. The three-dimensional GHZ state of this scheme was used for the key transfer and the detection of the decoy state particles, and the distributor did not need to send the particles that she holds to the key reconstruction during the detection of the decoy state particles and the reconstruction of the key. This protocol is more practical, secure, and quantum resource efficient compared with similar processes. 

## Figures and Tables

**Figure 1 entropy-25-00265-f001:**
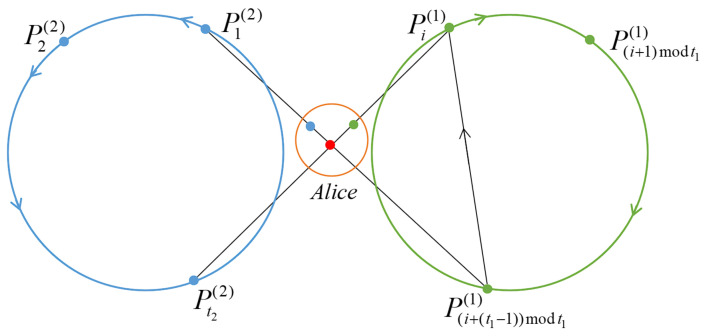
Structure diagram of the quantum network for this scheme.

**Figure 2 entropy-25-00265-f002:**
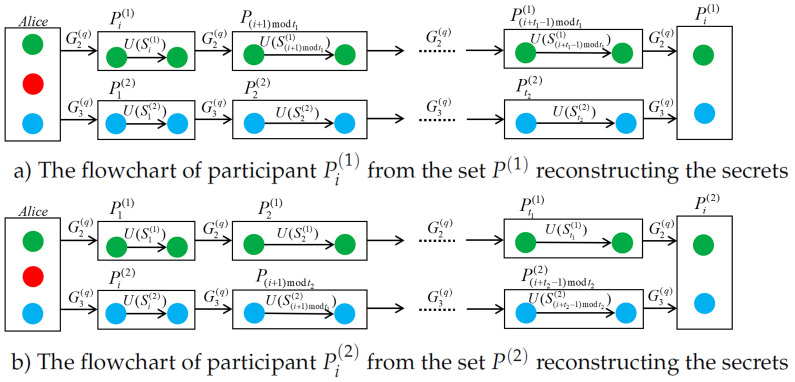
The process of the information particles transferring.

**Table 1 entropy-25-00265-t001:** Correlation of two-sided measurements of $|{\mathrm{GHZ}}_{000}\rangle$ and $|{\mathrm{GHZ}}_{100}\rangle$.

	Measurements of $P_{i}^{\left(1\right)}$			
**Alice**	$|{\mathrm{GHZ}}_{000}\rangle$		$|{\mathrm{GHZ}}_{100}\rangle$	
|+x〉	|+x〉	|+x〉	|+x〉	|−x〉
|+x〉	|−x〉	|−x〉	|−x〉	|+x〉
|+x〉	|+y〉	|−y〉	|+y〉	|+y〉
|+x〉	|−y〉	|+y〉	|−y〉	|−y〉
|−x〉	|+x〉	|−x〉	|+x〉	|+x〉
|−x〉	|−x〉	|+x〉	|−x〉	|−x〉
|−x〉	|+y〉	|+y〉	|+y〉	|−y〉
|−x〉	|−y〉	|−y〉	|−y〉	|+y〉

**Table 2 entropy-25-00265-t002:** Comparisons among several kinds of multi-party QKA protocols.

	RP2015 [22]	YGWQZW2015 [26]	BLWLL2018 [16]	LYZ2021 [25]	Our Scheme
Number of participants reconstruction key	2	2	*k*	1	1
Information particle trajectories	Tree form	Tree form	Tree form	Single circle	Double circle
Information quantum states	GHZ state (with *t* particles)	GHZ state (with *t* particles)	GHZ state (with *t* particles)	Generalised Bell state (with two particles)	GHZ state (with three particles)
The dimension of information quantum states	2	*k*	*k*	*k*	2
Detection of quantum states	GHZ state (with *t* particles)	Single photon	GHZ state (with *t* particles)	Single photon	Three dimensions GHZ state
Number of measurements	t(m+L)	t(m+L)	t(m+L)	t(2m+l)	3(m+L)
Number of unitary operations	0	0	0	t+1	t+1
Hash function	N	N	N	Y	Y
Information efficiency	$\frac{m}{t(m+L)}$	$\frac{m}{t(m+L)}$	$\frac{m}{t(m+L)}$	$\frac{m}{t(2m+l)}$	$\frac{m}{3(m+L)}$

## Data Availability

Not applicable.

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
