# Peer review of "An Efficient Quantum Secret Sharing Scheme Based on Restricted Threshold Access Structure"

_entropy, 2023, doi:10.3390/e25020265_

Round 1

Reviewer 1 Report

In this manuscript, authors claimed that they construct a quantum secret sharing scheme built on a constrained (t, n) threshold access structure, where n is the number of participants and t is the threshold number of participants. It seems that this scheme has passed the security analysis, but there are major problems listed below that need to be clarified by the author.

1.     The proof of Lemma in Sec. 1 is not unclear

1)     The equation between lines 71 and 72 : Two quantum states are both $| GHZ_{000} \rangle$, which one is $ | GHZ_{100} \rangle $?

2)     Line 80: “Lemma 1 holds when $ |GHZ_{000} \rangle =|GHZ_{100} \rangle $” should be “Lemma 1 holds when $ |GHZ \rangle =|GHZ_{100} \rangle $”.

3)     The equation between lines 86 and 87 : the left and right of the first symbol “=” are the same; the second “$ |00 \rangle $” should be “$ |11 \rangle $”.

4)     The case of “$ |GHZ \rangle =|GHZ_{100} \rangle $” and the second and the third equations between lines 83 and 84 should be proved.

2.     The logic of the protocol and the use of symbols are confussing.

1)     In this protocol, only $n_1+ n_2$ or $t_1+ t_2$ participants participate the protocol, not $n$, how to construct a (t, n) threshold protocol?

2)     Line 133: “the participant $P_i^{(1)} (i \in {1, 2, · · · , n_1})$ from the set $P_1$.”, how is the range of $n_1$ determined?

3)     Line 149: Only “the participant $P_1^{(2)} $ does a phase shift of $U_3(S_1^{(2)}) $ …”, how about the participant $P_i^{(1)} $?

4)     Lines 150-151: “participants $P_i^{(1)} $ and $P_1^{(1)} $ sends … to participants $P_{i+1 mod t_1}^{(1)} $ and $P_1^{(2)} $ …” should be “participants $P_i^{(1)} $ and $P_1^{(2)} $ sends … to participants $P_{i+1 mod t_1}^{(1)} $ and $P_2^{(2)} $ …”?

5)     Lines 154-156: “The participants $P_{i+1 mod t_1}^{(1)}$ and $P_2^{(2)} $ do a phase …. Then, they send … to participant $P_{i+1 mod t_1}^{(1)} $ and participant $P_2^{(2)} $, respectively, …” They send particles to themselves?

6)     Line 212: “green ball … , red ball … , basketball …” should be “green ball … , red ball … , blue ball …”? But, what does yellow ball denote?

7)     “$U(S_n^{(1)}) $” should be “$U(S_{n_1}^{(1)})$” in Fig 2(b)?

3.     The correctness of this protocol still needs to be verified.

1)     The following property needs to be proved at first.

$ U_1 (a) \otimes U_2 (b) \otimes U_3 (c) | GHZ \rangle

= U_1 (b) \otimes U_2 (a) \otimes U_3 (c) | GHZ \rangle $

2)     From line 133 “the participant $P_i^{(1)} (i \in {1, 2, · · · , n_1})$ from the set $P_1$” and Figure 2, it can be seen that only $n_1+ n_2$ participants do a phase shift but not $t_1+ t_2$. Therefore, the equation in lines 228-229 cannot prove the correctness of the protocol.

4.     The security of this protocol also exists some problems. Although Alice messed up the order, the order information was told the participants when the particles were transferred. Therefore, scrambling is invalid. For example, Eve can conduct Man-in-the-middle Attack.

In addition, there some other errors in this manuscript.

1.     Line 220: “Correctness of this protocal” should be “protocol”;

2.     Ref. [16] is inconsistent with the text “BLWLL2018[16]”;

3.     Ref. [25] should be “Li F L, Yan J Y, Zhu S X, …”.

Author Response

Please see the attachment “Respond 1.docx”.

Reviewer 2 Report

Quantum secret sharing is an important research branch of quantum cryptography. This paper constructs a quantum secret sharing scheme based on restricted threshold access structure, in which the author makes use of an internal quantitative relationship among three particles for GHZ states by phase-shifting operation, and finally gets the secret sharing. Compared with similar schemes, this paper uses less quantum resources, and the efficiency of this scheme is relatively high. The idea is novel and the paper is worth publishing after some revisions as follows.

1. In the introduction, several numbers "n" need to be modified for mathematical context.

2. Page 2, line 34, the sentence "...and an angle a shifting operations performed" should be "...and an angle a shifting operations performed by… ".

3. Page 4, line 19, the sentence "…then shares H() with the participants  from the set P" should be "…then shares H(S) with the participants from the set P".

4. In Figure 1, it is best for the author to add Alice to the diagram as well.

5. Page 8, line 1 from the bottom, the author adds a note to s, because I think the second representation of s is used at this time, that is,s = (s-1,s-2,... s-m).

6. I have also carefully reviewed the data in Table 2 and I think the authors should refine the description of indicators and figures in Table 2. For example, "2 dimensions" can simplify to "2", etc.

Author Response

Thank you for your valuable comments. Please see the attachment “Respond 1.docx”.

Round 2

Reviewer 1 Report

The revised version has improved significantly compared with the previous version, but there are still some problems to be clarified, as listed below.

1.     Is the protocol proposed in this manuscript designated a fixed set of t-1 participants to share the secret information with the distributor Alice, and only these t-1participants participate the protocol, rather than *any* t of the n participants can get the secret information as in the prior art protocols ? Please made this clear in the manuscript.

2.     Some mathematical symbols are still confusing in this reversion.  For example, in “$P=\{ P_1, P_2, …,  P_n\}$”, $P_1$ and $P_2$ represent two participants, but thereafter all occurrences are defined as two subsets of $P$ .

3.     The goal of proof is to prove formula (2a), instead of "Because $(U_1 (a) \Otimes I \Otimes I) (U_1 (b) \Otimes I \Otimes I)… $". Please make correct presentation.
